# Identification of Environmental Determinants Involved in the Distribution of *Burkholderia pseudomallei* in Southeast Asia using MaxEnt software

**Jose Francis V. Abrantes**[1,2]*, **Zenn Ashley P. Cariño**[1¤a], **Hozeo Luis S. Mercado**[1¤b], **Fatima N. Vicencio**[1¤c], **Gio Ray S. Sosa**[1¤b], **Miguel Angelo M. Habaña**[1¤d], **Nikki Heherson A. Dagamac**[1,2]

1 Department of Biological Sciences, College of Science, University of Santo Tomas, Manila, Philippines,
2 The Graduate School, University of Santo Tomas, Manila, Philippines

¤a Current address: Faculty of Medicine and Surgery, University of Santo Tomas, Manila, Philippines
¤b Current address: University of the East Ramon Magsaysay Memorial (UERM) Medical Center, Quezon City, Philippines
¤c Current address: Chinese General Hospital Colleges, Manila, Philippines
¤d Current address: Far Eastern University (FEU)-Nicanor Reyes Memorial Foundation (NRMF) Medical Center, Quezon City, Philippines
* jvabrantes@ust.edu.ph

**Data Availability Statement:** Publicly available datasets were analyzed in this study. The list of occurrence raw data used for the distribution of Bp

## Abstract

*Burkholderia pseudomallei* (Bp), causing melioidosis, is becoming a major global public health concern. It is highly endemic in Southeast Asia (SEA) and Northern Australia and is persisting beyond the established areas of endemicity. This study aimed to determine the environmental variables that would predict the most suitable ecological niche for this pathogenic bacterium in SEA by maximum entropy (MaxEnt) modeling. Systematic review and meta-analysis of data for melioidosis were obtained from public databases such as PubMed, Harmonized World Soil (HWSD) and WorldClim. The potential map showing the environmental layers was processed by ArcGIS, and the prediction for the probability of habitat suitability using MaxEnt software (version 3·4·4) and ENMeval R-based modeling tools was utilized to generate the distribution map with the best-fit model. Both bioclimatic and edaphic predictors were found to be the most important niche-determining environmental variables affecting the geographical distribution of Bp. The highest probability of suitability was predicted in areas with mean temperature of the wettest quarter at ≥26°C, annual precipitation of <2300 mm and Acrisol soil type. Combining those significantly influential variables, our predictive modeling generated a potential distribution map showing the concentration of areas and its location names with high suitability for Bp presence. The predicted distribution of Bp is extensive in the mainland part of SEA. This can be used to draw appropriate measures to safeguard public health and address the true disease burden of melioidosis in the region under the current climate scenario.

is available at Zenodo (https://doi.org/10.5281/zenodo.11507319).

**Funding:** The authors declare that no external funding was received for the conduct of this study. However, partial financial support for the article processing charges was provided by the University of Santo Tomas, Philippines. The funders had no role in the study design, data collection and analysis, decision to publish or preparation of the manuscript.

**Competing interests:** The authors have declared that no competing interests exist.

## Author summary

Bp causes melioidosis and the incidence of melioidosis is significantly associated with high endemicity of this pathogenic bacterium in tropical Southeast Asia and Northern Australia. There are usually seasonal outbreaks of the disease due to human encounters with the pathogen in severe weather events since it is largely influenced by environmental variables. Using MaxEnt species distribution modeling to predict its occurrence, the environmental variables identified in this study were both bioclimatic and soil predictors (temperature, annual precipitation, and Acrisol-rich soil environment) that strongly influence the geographical distribution of Bp. Combining those significantly influential variables, our predictive modeling generated a potential distribution map showing the concentration of areas and location names with high suitability for Bp presence. This predictive map showed extensive occurrence in the mainland of SEA and should be prioritized for public health interventions and protect those individuals with additional health risk factors in acquiring melioidosis. In an ongoing climate change condition, there is a need to predict further future Bp geographical distribution under different climate change scenarios within the areas of high endemicity and beyond. This is intended to establish early public health warning systems before the disease becomes a global public health problem.

## Introduction

*Burkholderia pseudomallei* (Bp) causes melioidosis and is becoming a major global public health concern [1]. This infectious disease is potentially fatal to humans and is often referred to as a great mimicker of various diseases [2,3]. It presents a wide spectrum of clinical manifestations ranging from latent or localized infections to disseminated lethal diseases including tuberculosis-like illness [4]. Treatment for this disease presents a serious concern because Bp has intrinsic resistance to many antimicrobial agents [5]. The etiology is a facultatively intracellular Gram-negative bacterium. It is likewise, a free-living environmental saprophyte found persisting in soil and surface water in both endemic tropical and nonendemic temperate regions [6]. Bp can be transmitted by percutaneous inoculation through skin abrasion, inhalation, or ingestion. An estimate of the global burden of melioidosis and Bp has been found to be surprisingly widespread using a modeling approach. Projected human cases are predicted to be around 165,000 of whom 89,000 are expected to die yearly [7]. Using different infection modeling schemes, additional health risk factors were identified with the incidence rate of melioidosis in a highly endemic area. These include mobility change of residents, working males between 45 to 59 years old and more importantly, diabetic individuals [8,9]. Individuals with chronic renal and liver diseases were also reported to be susceptible to melioidosis [10].

Since Bp is autochthonous to soil and surface water, changes in climatic processes and the potential exposure of susceptible individuals in endemic regions are expected to modify melioidosis epidemiology and can have significant impact on the intensity of seasonal occurrence of outbreaks. Early on, observation on clustering of cases was associated with total rainfall [11]. A rise in the dew point, cloud cover, maximum temperature and groundwater have been implicated as additional environmental factors in the uptick of incidence rate of infection [12]. Descriptions on the aberration in climate and weather events were found to increase melioidosis cases after storms in Taiwan and cyclones in Northern Australia [13]. Significant rise of disease burden among adults and children in Cambodia was observed during rainy days with high humidity and maximum wind speed while low visibility and windy conditions

with humidity were noted in Laos [14]. These changes in weather pattern could influence and control the distribution, growth and persistence of Bp in soil ecosystems. This is coupled with anthropogenic activities that could further drive to a new spatial and temporal disease outbreak. The soil characteristics of porous Acrisol, Anthrosol and Luvisol types were found to favor dissemination of this pathogenic bacterium. Increased soil porosity as a result of heavy rains and flooding initiates the movement of the organism from latently quiescent state in the soil rising through the water surface, recrudescing, and thus exposing subsequently susceptible animals and human populations [15,7,16,10].

The environmental factors involved must be determined and assessed in the distribution of Bp to account for the optimal conditions potentially driving seasonal outbreaks. Changes in temperature, global precipitation patterns, and severe weather events can lead to prolonged seasonal output of Bp with profound public health implications due to the increased intensity of melioidosis outbreaks [13,10]. Early public health interventions are needed to document the current disease burden which is misdiagnosed or underreported in highly endemic areas of Southeast Asia (SEA). Prediction by species distribution and ecological niche modeling can present the highest probability of occurrence of Bp that can be linked to projected incidence rate of melioidosis in this region. We utilized an open-source MaxEnt software to identify and select the environmental determinants most suitable for the distribution of Bp in this highly endemic area [17].

## Methodology

### Data collection

Preferred reporting items for systematic reviews and meta-analyses (PRISMA) guidelines were employed to collect reports and cases of Bp from different sources since its inception from October 17, 2022, until April 28, 2023 (Fig 1). The melioidosis database (www.melioidosis. info) was used to collect records. Search for reports in PubMed database was done using combinations of keywords and MESH terms, namely, "*Burkholderia pseudomallei*", "melioidosis", "Southeast Asia", and "sampling". One thousand forty-two records were initially identified from these databases. Before screening, two hundred thirteen duplicate records were removed. The remaining eight hundred twenty-nine records were then assessed for eligibility by 1) selecting records obtained from SEA, 2) selecting only records obtained from environmental soil sampling, and 3) selecting records with precise geographic coordinates. The accuracy of all the one hundred forty-four geographic coordinates used for this modeling study was verified by conducting an initial data check. In accordance with the approach used by Limbo-Dizon and co-workers [18], the spThin package in R software using the spatial filtering approach was used to reduce spatial autocorrelation, which is the tendency of closer locations to be more similar than those further apart. In this procedure, the presence records were randomly selected according to a minimum nearest neighbor distance ≥10 km between each locality [19]. This approach resulted in ninety-nine geographical coordinates from the initial one hundred forty-four coordinates. The procedure was repeated ten times to produce a spatially filtered data set. These filtered geographical coordinates were saved as a comma-delimited (CSV) file type. These files are now publicly available at Zenodo (https://doi.org/10.5281/zenodo.11507319).

### Acquisition of environmental layers

Factors identified in the study were hypothesized and considered to contribute to the presence of Bp in an area. Variables included are (1) soil characteristics from the Harmonized World Soil Database (HWSD) (http://www.iiasa.ac.at) and (2) nineteen bioclimatic variables from the

**Fig 1. PRISMA flow diagram.** Depiction of core procedures used in this review process such as the identification of records from different sources (databases and registers), removal of records, screening, and the inclusion of the remaining reports.

WorldClim database (www.worldclim.org). Map layers come from the USGS public domain. All the downloaded environmental layers from HWSD and WorldClim databases were in GeoTIFF format. The ArcGIS software converted all the environmental layers into an ASCII extension. Using ENMtools in R software, highly correlated variables >0.7 were excluded from the study resulting in seven environmental variables (Table 1) from the initial nineteen

**Table 1. Percent Contribution and Permutation Importance of Selected Environmental Variables for the Suitability of Bp in SEA Countries.**

| Environmental Variables | Percent contribution | Permutation importance |
|---|---|---|
| Mean Temperature of Wettest Quarter (**Bio8**) | 67.9 | 46.4 |
| Soil type (**Biocateg1**) | 13.1 | 15.9 |
| Annual Precipitation (**Bio12**) | 9.2 | 18.3 |
| Mean Temperature of Driest Quarter (**Bio9**) | 6.6 | 11 |
| Precipitation of Wettest Quarter (**Bio16**) | 3.1 | 7.1 |
| Annual Temperature Range (**Bio7**) | 0.1 | 1.1 |
| Precipitation of Warmest Quarter (**Bio18**) | 0.1 | 0.2 |

variables [20]. This was done to 1) avoid overfitting, 2) obtain accurate response curves, and 3) prevent the model from being adjusted excessively to the data [21].

## Species distribution modeling

Maximum Entropy Model (MaxEnt Version 3·4·4) from https://biodiversityinformatics. amnh.org/opensource/maxent/ was used to predict the potential environmental niche of Bp. MaxEnt predicts the distribution (geographic range) of a given species by using presence-only data and finding the distribution that has maximum entropy (i.e. is the most geographically uniform) while being restricted to environmental constraints [22]. The software can handle continuous and categorical predictor variables while also automatically incorporating variable interactions. Specifically, we used various characteristics using environmental variables such as topographic, edaphic, and bioclimatic to restrict the geographic distribution. In this open-source software, an evaluation of the species' habitat suitability ranges from 0 (least suitable) to 1 (highest suitability) [23,24].

Prior to the final model run, the dataset was subjected to an analysis using ENMeval R-based evaluation package [25]. This data quality filtering was included to optimize model complexity in order to balance the goodness of fit and predictive ability avoiding any possibility of overfitting [26]. The fine-tuned setting produced from the ENMeval analysis (method = randomkfold, kfold = 10) recommended the adjustment of a regularization multiplier (RM) of 3 and a feature type (FT) of a linear feature. To determine the significance of each environmental variable and its relative contribution to the model prediction "*Create a response curve*" and "*Do jackknife test to measure variable importance*" were ticked respectively. Model performance was assessed using the generated Area under the ROC (receiver-operator characteristic) curve (AUC) which ranges from 0 to 1 and is a metric for model performance [27]. The MaxEnt ASCII output file format was imported into the ArcMap 10·8 software for better visualization. For the specific classification of classes, the map was divided using an equal five-part interval. Each species' habitat suitability was then translated from the five-part interval into five categories namely, very low suitability, low suitability, moderate suitability, high suitability, and very high suitability.

## Results

A total of twenty-six reports resulted after the systematic review and meta-analysis of pooled data. This was then used in the study to collect one hundred forty-four geographic coordinates for the occurrence records. Ninety-nine occurrence points were found in geographical ranges in SEA for Bp after spatial filtering. The projected model shows a potential distribution pattern focusing on the mainland part of the region (Fig 2).

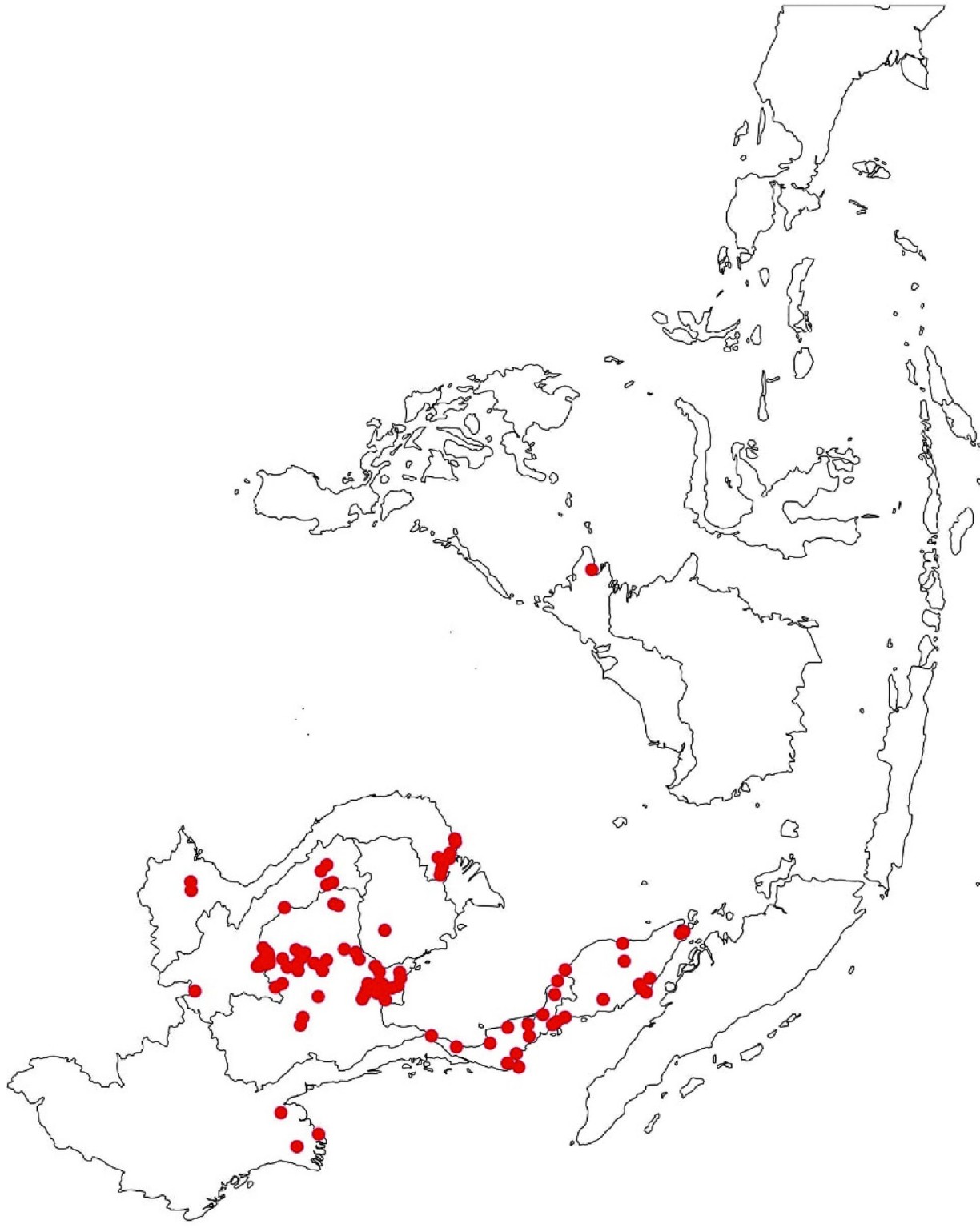

**Fig 2. Occurrence points found in geographical ranges in SEA for Bp (Plotted in red).** The base map layer was obtained from the publicly available database of Global Administrative Area (GADM) under the license of https://gadm.org/license.html. The figure was created using QGIS (www.qgis.org).

To consider the environmental variables that contributed to the presence of the bacterium, six bioclimatic and one edaphic (soil) variables were identified while maintaining a high-level quality of model fitting. Our species distribution modeling revealed several significant predictors that could possibly affect the species' projected regional distribution in SEA (S1 Table). Both bioclimatic and edaphic predictors showed strong influence on Bp habitat suitability. The model identified the percent contribution and permutation importance of these seven variables involved. On the basis of using the three algorithm-specific indicators of variable importance, three of the seven bioclimatic and soil characteristics were among the most contributory predictors. Bio8 (mean temperature) contributed 67·9% probability of suitability, followed by Biocateg1 (soil type) and then Bio12 (annual precipitation). With regards to permutation importance, the model identified Bio8 as the most important, preceded by Bio12 and then Biocateg1. These contributory variables accounted for approximately 80·6% of the overall permutation importance and collectively add to the formulation of our model.

To evaluate our model for predictive accuracy, we employed the commonly used baseline metric AUC curve [26], which generated a value of 0.882 indicating a strong model efficiency (Fig 3). The AUC is an important index that provides a single measure of model performance independent of any particular threshold [28,17]. The value of AUC ranges from 0 to 1, values ≤ 0.5 indicate that the model's performance is no better than random, while values closer to 1 indicate that the model's performance is more reliable [29]. Furthermore, we subjected the occurrence data against these two significant environmental predictors (Bio8 and

**Fig 3. The illustration graph of the area under the curve (AUC) value (0.882) indicates a strong predictive model efficiency.**

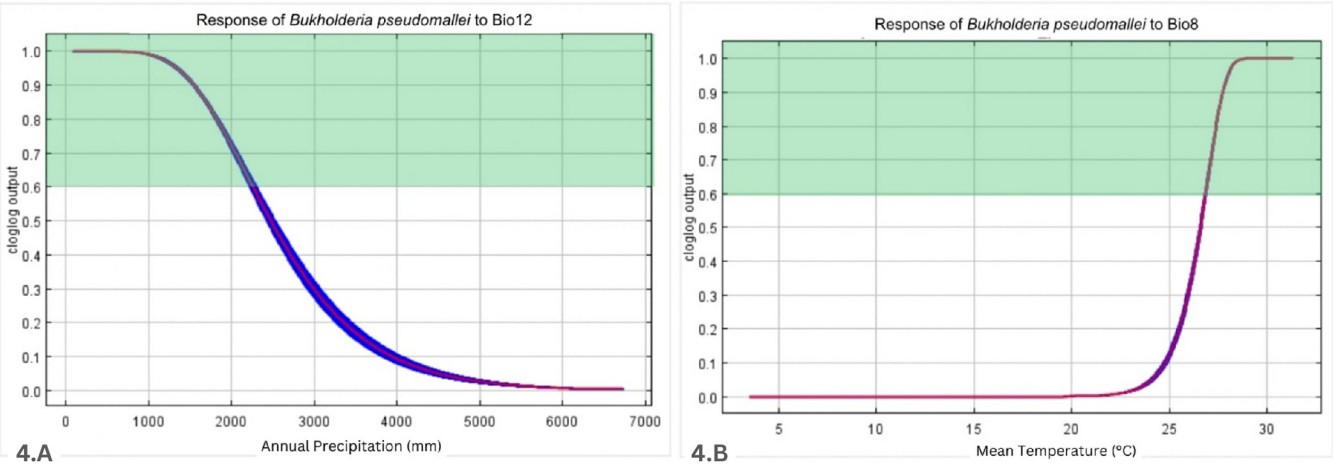

**Fig 4. Illustration graphs of response curve for Bp.** (A) annual precipitation (Bio12) and (B) mean temperature of wettest quarter (Bio8). Values within the highlighted region (green) contribute to high and very high suitability.

Bio12) to evaluate the plausibility of our model depicted by response curves (Fig 4). In the generated response curves, Bio8 showed increasing cloglog (complimentary log-log) outputs as this variable increases, wherein a temperature of above 26°C contributed to either high ($\geq$0.6) or very high suitability ($\geq$0.8) (Fig 4B). On the other hand, the variable annual precipitation showed a decreasing cloglog output as the variable increases, where Bio12 of <2300 mm contributed to either high ($\geq$0.6) or very high suitability ($\geq$0.8) (Fig 4A). As for soil characteristics (Biocateg1) from HWSD, only Acrisol contributed to either high or very high suitability as it generated a cloglog output of >0.6 (Fig 5). Gleysol and Luvisol generated a cloglog output of <0.5. The remaining soil types reflected a cloglog output of <0.35.

Based on Bp's current species occurrence data, our predictive model generated a distribution map showing the areas in SEA that have the suitable conditions for the bacterial pathogen (Fig 6). As shown in S1 Table, several areas of nine out of the eleven SEA countries exhibited high to very high suitability for Bp. This is also indicative of high distribution of the species in that respective area. Brunei and Singapore exhibited low suitability to Bp. The estimated places that have either high or very high suitability based on the generated map (Fig 6) are also listed in S1 Table. The number of places mentioned per country does not necessarily reflect the land area suitable to harboring Bp. This is merely an estimation in respect to each country's regional, provincial, or district borders. The names of geographic locations are enumerated for reference to initiate public health intervention and decision-making.

## Discussion

Spatial and meta-analysis of twenty-six pooled occurrence data revealed three important environmental variables (Bio8, Biocateg1, Bio12) that predict ecological niche suitability and distribution of Bp in SEA. The contribution of geographical range-determining variables corresponded to the generation of our projected distribution map in areas showing its extensive potential distribution by MaxEnt modeling (Fig 6). The combination of these environmental predictors in our model is consistent with previous studies and observations [7,12,13,16,10].

### Model limitations

The usage of robust machine learning predictions such as the MaxEnt algorithm has been increasing rapidly in the span of the last decade. In fact, for the Southeast Asian region alone,

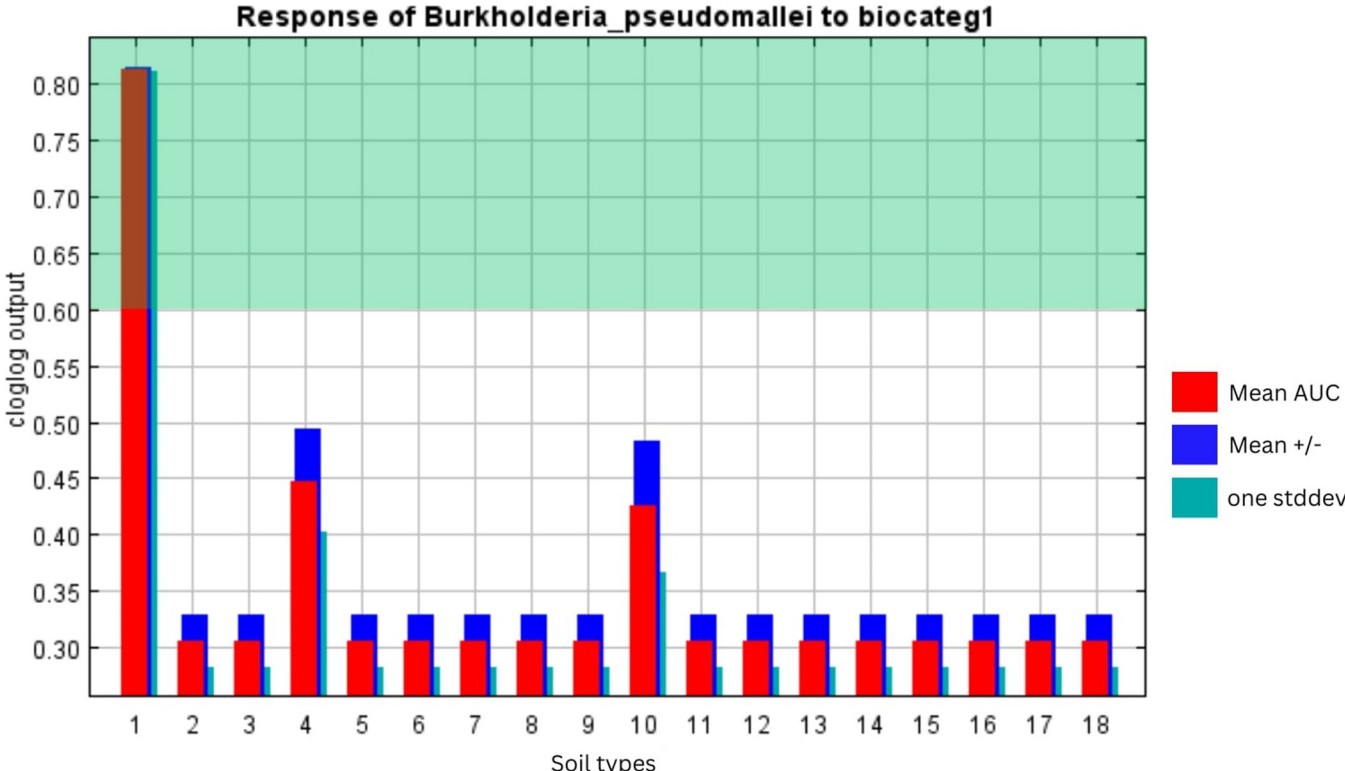

**Fig 5. Illustration graph of response of Bp to Soil Type (Biocateg1).** Values within the highlighted region (green) contribute to high (≥0.6) to very high suitability (≥0.8). (1) Acrisol, (2) Cambisol, (3) Leptosol, (4) Gleysol, (5) Fluvisol, (6) Luvisol, (7) Lixisol, (8) Nitisol, (9) Andosol, (10) Solonchak, (11) Arenosol, (12) Vertisol, (13) Plinthosol, (14) Ferrosol, (15) Calcisol, (16) Histosol, (17) Podzol, and (18) Water Bodies.

this algorithm has been using correlation mechanisms to predict suitable environmental habitats for elusive animals [30], plants [31], soil protist [32], phytopathogens [33], and even cases of neglected parasitic diseases [34]. Despite such promising potentialities that the predictive power of a model can generate, it is important to note that such models are heavily reliant on excellent types of data being employed for the model. Hence, besides the number of recorded occurrences for the Bp, we are limited on which environmental predictors to use. Therefore, for this study, data that are easily available and transformed into meaningful spatial analysis are relatively considered. As such, despite the knowledge of complex mechanisms that can transpire in melioidosis, like land use, anthropogenic activities, and climate change, we have first limited the models to data that are much correlative in nature (bioclimatic, edaphic, topographic), like what has been done in the aforementioned studies. This is because abiotic factors such as those that were selected for this study better support environmental assumptions about melioidosis [35,36]. For a more mechanistic approach, it is highly suggested that we consider factors that are highly attributed for melioidosis and such theme warrants further investigations once reliant data becomes available for the region.

## Important Predictors for Bp Precipitation

Our results showed both bioclimatic and edaphic predictors are equally significant in our model demonstrating the hydroclimatic influences on Bp-receptive soils where its distribution is directly implicated with the rise of melioidosis. Seasonal foci of disease outbreaks have long been associated with heavy rainfall events [11]. Subsequent bacterial output from groundwater

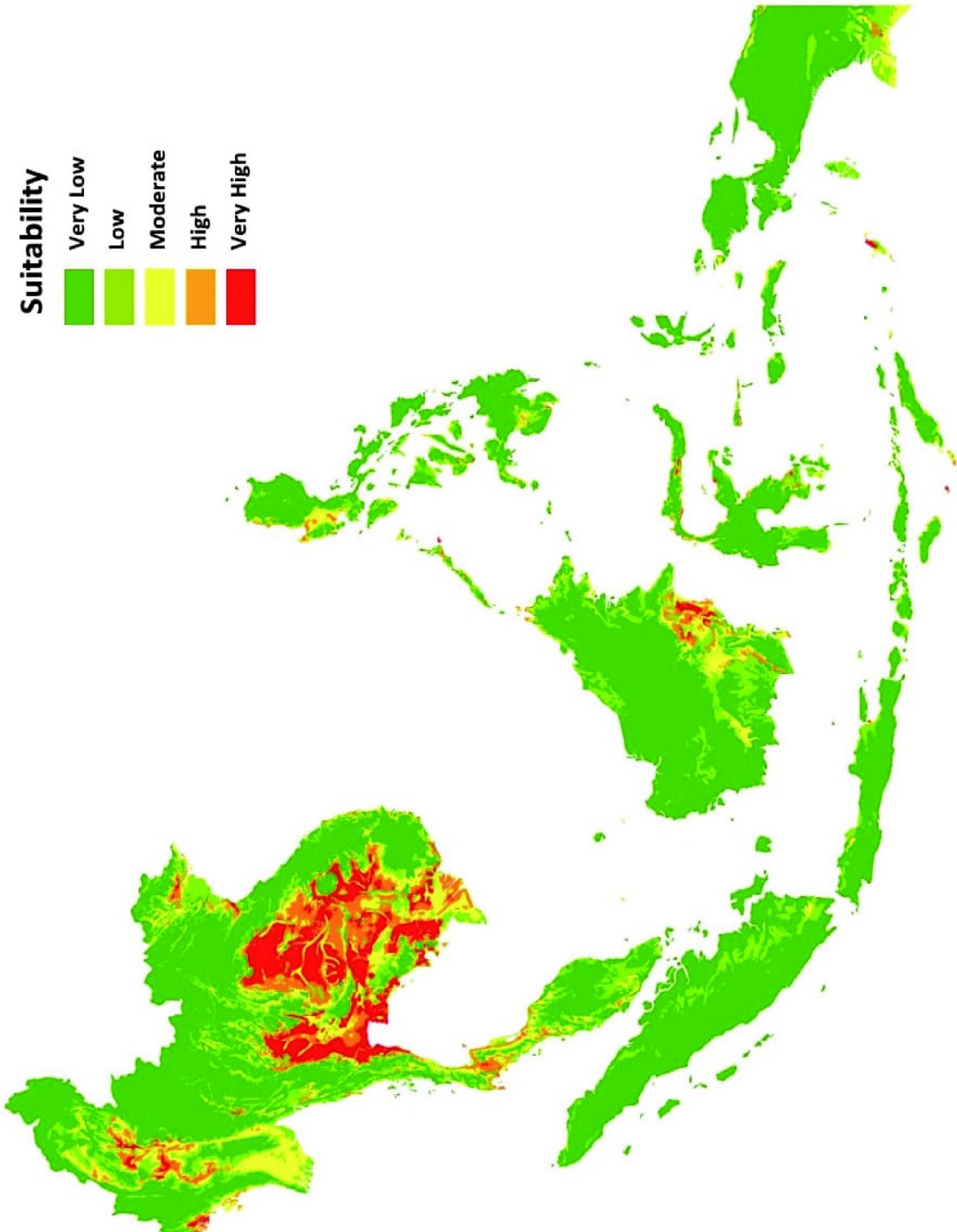

**Fig 6. Generated MaxEnt model predicting the possible distribution of Bp in SEA.** Dark green = very low suitability (0–0.19); light green = low suitability (0.2–0.39); yellow = moderate suitability (0.4–0.59); orange = high suitability (0.6–0.79) and red = very high suitability (0.8–1). The base map layer was obtained from the publicly available database of Global Administrative Area (GADM) under the license of https://gadm.org/license.html. The figure was created using ARCGIS.

seeps after rain or flood, exacerbated by high humidity and windy conditions, were postulated to bring about the shift towards airborne mode of transmission. This leads to a more serious form of illness and contribute to clinical clustering [11,14]. The effects of Bio12 were less significant to the model performance than the other variables (Bio8, Biocateg1) because of low values of percent contribution and permutation importance. This can possibly be attributed to changes in precipitation pattern and monsoonal weather conditions [13,9]. As shown in Fig 4A, an annual precipitation of <2300 mm contributes to high to very high suitability for the bacterial pathogen. It was observed that Bp was more common in Myanmar, Thailand, Laos, Cambodia, and Vietnam during the Southwest monsoon with an annual precipitation of ~1000 to 2800 mm [37] and similarly depicted by our model in Figs 3 and 6. This can also be associated with melioidosis cases peaking annually during the rainy season (May–October) in both Laos and Cambodia whereby the microorganism accumulating on surface soils occurs as a result of increasing water tables brought only by incremental rainfall [38,14]. This follows the trend observed in Fig 4A where there is less possibility for the bacterial presence in places with annual precipitation of over 2300 mm.

## Soil type

The contribution of Biocateg1 in our model cannot be ignored with its potential habitat since the bacterium is intrinsically a soil dweller. Acrisol type contributed either high or very high suitability for Bp (see Fig 5). Our model has identified acidic, clay rich Acrisol consistent with the study of Goodrick [16] and Pongmala [39] and their co-workers. Other preceding studies accounted for not only Acrisol but also Anthrosol [7] or Luvisol type [15] while only Anthrosol type was associated by Birnie and co-workers [10]. The discrepancies in predicting soil type in these different reports may be due to various detection techniques and different modeling schemes used. Nevertheless, the implication of primarily associating Acrisol type or with other soil types is with the infection risk for melioidosis since many individuals are working for their livelihood in the tropical rural areas. Most often, they are involved in agriculture or domesticated animal industry. This is exemplified by workers tilling soil in rice paddy fields barefoot and are constantly exposed to the soil environment [15,4]. These rural areas in SEA are the ones lacking adequate medical services that missed out on the definitive diagnosis of melioidosis. Thus, we presently do not have the true picture of disease burden in this region. This is exacerbated by various anthropogenic activities on a shift in rural land use.

## Temperature

Our predictive model suggests temperature (≥26˚C) is the most influential of all the environmental predictors identified (Bio8) and has the greatest impact of more than sixty-seven (67·9%) percent contribution and forty-six (46·4%) in terms of permutation importance. Referring to the term used in this software, the mean temperature of the wettest quarter is the index that approximates the mean temperatures that prevail during the wettest season [40]. In an earlier study by Liu and co-workers [38], the increasing cases of melioidosis were noted from July to October, and January having a mean temperature of between 27.7˚C ± 0.7˚C. Similarly, SEA countries experiencing greater amounts of rainfall during the Southwest monsoon season and increasing ambient temperature were observed to have a higher risk for melioidosis [12,7]. Investigation of the physiologic characteristics of Bp at a wider range of temperatures demonstrated the bacterium can sustain growth, motility, biofilm formation, and resistance to oxidative stress at 37˚C [41]. This was further supported by findings of the optimal temperature for the growth of Bp in soil media were between 37˚C and 42˚C [42]. All these previous studies coincided with our findings shown in Fig 4B that areas with higher

temperature ($\geq 26\degree$C) have an increased likelihood for the bacterial presence. With the present onset of global warming, the ecology of this microorganism is likely to change the disease dynamics and epidemic potential amid sea surface and ambient temperature increase. This has been clearly anticipated by many studies [11,12,7,13,10].

## Implications for Bp epidemiology

Overall, the predictive model suggested that the potential highest suitability of Bp presence peaks during the Southwest monsoon, where there is usually heavy rainfall with the sea surface and ambient temperatures at more than 26˚C in Acrisol-laden areas are the conditions most suitable for this bacterial pathogen plotted in our MaxEnt distribution model (Fig 6). Most areas, that exhibited high to very high suitability, are clustered around mainland SEA (Cambodia, Laos, Malaysia, Myanmar, Thailand, and Vietnam). Some areas in East Timor, Indonesia, and Philippines also exhibited high to very high suitability. Brunei and Singapore did not show high suitability for the bacterium. Such information can now be targeted to document the seasonal burden of melioidosis in terms of morbidity and mortality and draw public health guidelines and safeguards, including those with additional health risk factors in acquiring melioidosis [8]. This will then reduce the disparity of true number of cases in these areas with the numbers predicted by many models. Our study suggests rising temperature, annual precipitation and geographic presence of Acrisol soil are important environmental-determining drivers for the distribution of Bp in SEA under the current climate scenario. The best-fitting model revealed its predicted occurrence is more influenced by bioclimatic variables than edaphic factors that will change melioidosis epidemiology due to the ongoing climate change. As such, there is a need to predict further future Bp geographical distribution within the areas of high endemicity and beyond. Hence, this regional modeling that scopes only the tropical to subtropical climates of the Southeast Asian region is already a big epidemiological step in utilizing innovative technologies such as machine learning to establish early public health warning systems before the disease becomes another global public health problem.

## Supporting information

**S1 Table. SEA Countries with High to Very High suitability to Bp.**
(DOCX)

## Acknowledgments

We expressed our unwavering gratitude to the Office of the College of Science, most especially to Dr. Reuel M. Bennett and Dr. Richard Thomas B. Pavia, Jr. as Department Chairs of the Biological Sciences of the University of Santo Tomas. All authors are indebted to the support of Office of the Vice Rector for Research and Innovation and the Office of the Graduate School.

## Author Contributions

**Conceptualization:** Jose Francis V. Abrantes.

**Data curation:** Jose Francis V. Abrantes, Zenn Ashley P. Cariño, Hozeo Luis S. Mercado, Fatima N. Vicencio, Gio Ray S. Sosa, Miguel Angelo M. Habaña, Nikki Heherson A. Dagamac.

**Formal analysis:** Jose Francis V. Abrantes, Zenn Ashley P. Cariño, Hozeo Luis S. Mercado, Fatima N. Vicencio, Gio Ray S. Sosa, Miguel Angelo M. Habaña, Nikki Heherson A. Dagamac.

**Investigation:** Zenn Ashley P. Cariño, Hozeo Luis S. Mercado, Fatima N. Vicencio, Gio Ray S. Sosa, Miguel Angelo M. Habaña.

**Methodology:** Nikki Heherson A. Dagamac.

**Project administration:** Jose Francis V. Abrantes.

**Resources:** Nikki Heherson A. Dagamac.

**Software:** Nikki Heherson A. Dagamac.

**Supervision:** Jose Francis V. Abrantes, Nikki Heherson A. Dagamac.

**Validation:** Jose Francis V. Abrantes, Zenn Ashley P. Cariño, Hozeo Luis S. Mercado, Fatima N. Vicencio, Gio Ray S. Sosa, Miguel Angelo M. Habaña, Nikki Heherson A. Dagamac.

**Visualization:** Jose Francis V. Abrantes, Zenn Ashley P. Cariño, Hozeo Luis S. Mercado, Fatima N. Vicencio, Gio Ray S. Sosa, Miguel Angelo M. Habaña.

**Writing – original draft:** Jose Francis V. Abrantes, Zenn Ashley P. Cariño, Hozeo Luis S. Mercado, Fatima N. Vicencio, Gio Ray S. Sosa, Miguel Angelo M. Habaña.

**Writing – review & editing:** Jose Francis V. Abrantes, Nikki Heherson A. Dagamac.

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
