## [Decision Letter · Decision Letter 0]

6 May 2024

Dear Dr Jose Francis Villarante Abrantes,

Thank you very much for submitting your manuscript, "Identification of Environmental Determinants Involved in the Distribution of Burkholderia pseudomallei in Southeast Asia using MaxEnt software" for consideration at PLOS Neglected Tropical Diseases. As with all papers reviewed by the journal, your manuscript was reviewed by members of the editorial board and by several independent reviewers. In light of the reviews below this email, we would like to invite the resubmission of a significantly revised version that takes into account the reviewers' comments. 

We cannot make any decision about publication until we have seen the revised manuscript and your response to the reviewers' comments. Your revised manuscript is also likely to be sent to reviewers for further evaluation.

Important additional instructions are given below your reviewer's comments.

Sincerely,

Dawit Gebeyehu Getachew, MPH

Guest Editor

Stuart Blacksell

Section Editor

Dear author, Thank you for submitting your manuscript titled ''Identification of Environmental Determinants Involved in the Distribution of Burkholderia pseudomallei in Southeast Asia Using MaxEnt Software''.

The editors have evaluated your manuscript and review report from multiple independent reviewers. But, before passing a decision on the acceptability of the manuscript, a revised form of your manuscript, together with a highlighted or changed manuscript, should be submitted.

Please also add a point-by-point response to the reviewer's comment and question.

Dawit Getachew 

PLOS Neglected Tropical Diseases (Gust Editor)

Reviewer's Responses to Questions

**Key Review Criteria Required for Acceptance?**

**Methods**

-Are the objectives of the study clearly articulated with a clear testable hypothesis stated?

-Is the study design appropriate to address the stated objectives?

-Is the population clearly described and appropriate for the hypothesis being tested?

-Is the sample size sufficient to ensure adequate power to address the hypothesis being tested?

-Were correct statistical analysis used to support conclusions?

-Are there concerns about ethical or regulatory requirements being met?

Reviewer #1: Yes

Reviewer #2: Yes, the hypothesis is understandable. Methods could be greatly improved. As a reader, I was quite confused as important terms are not sufficiently defined in the methods section. AT the same time, large parts of the methods text could be summarized more succinctly. The authors copy/paste command/settings :Lines 198-201 for previously published tools.

I also suspect that an entire figure was accidentally excluded or mis-referenced. In Line 173, the authors explain how a subset of the bioclimatic variables were excluded, but the figure referenced is a map. The authors miss an opportunity to clearly communicate how the tools were used. Each was previously published and open-source, but sentences like Line 186 to 188 are not clearly stated, and technically confusing.

Reviewer #3: (No Response)

**Results**

-Does the analysis presented match the analysis plan?

-Are the results clearly and completely presented?

-Are the figures (Tables, Images) of sufficient quality for clarity?

Reviewer #1: Yes

Reviewer #2: From what I can tell, yes, the analysis matches, but the authors again miss a big opportunity to clearly communicate their findings, instead of relying on overly technical vocabulary and jargon. The figure legends are frustratingly brief and at times confusing (Fig 3). Alone, the figure legend is unable to summarize the what the figure communicates (is this the final species distribution model being tested?). The results section is an opportunity for the authors to describe the env factors that did not appear to contribute to their assessment of habitat suitability? The other factors are ignored and seemingly deemed unimportant. The authors tell an incomplete story, and leave the reader frustrated by the lack of transparency. 

The authors must revisit each figure legend and consider writing sentences that help the reader understand what is captured. Figure 4 mentions "response curve for Bp to annual precipitation" which could lead to misunderstanding what is being tested, because the authors are predicting the usefulness of data here. I could be easily confused that the authors have expose B.pseudomallei to different amounts of rain.

Reviewer #3: (No Response)

**Conclusions**

-Are the conclusions supported by the data presented?

-Are the limitations of analysis clearly described?

-Do the authors discuss how these data can be helpful to advance our understanding of the topic under study?

-Is public health relevance addressed?

Reviewer #1: Yes

Reviewer #2: The authors miss an opportunity to clearly state their study findings. The first sentence repeats jargon and nomenclature that assumes the reader runs these analyses routinely (Bio8, Biocateg1, Bio12), instead of communicating how these are translated. As a scientist who had never run these analyses, the authors have missed an opportunity to translate the implications of their findings to the broad public health and scientific community. Lines 337 to 340 as standalone sentences if re-written, could belong in the ir results section, but are overly technical. How do the authors translate these permutation importance for public health? Again in lines360-362, the authors hold fast to code-words (Biocateg1, Bio8, Bio12) with zero context for the reader. It's not until Line 375-78 that the author reveal the code--this sentence should begin the discussion section, but instead this is nearly the final sentence of the report.

Reviewer #3: (No Response)

**Editorial and Data Presentation Modifications?**

Reviewer #1: (No Response)

Reviewer #2: (No Response)

Reviewer #3: (No Response)

**Summary and General Comments**

Reviewer #1: Overall, this is a very well written and interesting manuscript. 

Major: nil

Minor:

The abstract is well written. Does it need to have subheadings?

Line 128: prolonged

Line 173: should this be Supplementary Table 1 and not “Figure 2”

The use of Biocateg1, Bio8, or Bio12, can be difficult to follow while reading the manuscript. Could the authors please spell out what these factors are in the text instead. 

E.g. line 230: “Bio8 contributed 67·9% probability of suitability, followed by Biocateg1 and then Bio12”

Line 315: can the authors expand on why they think regions with precipitation >2300mm/year would have less B. pseudomallei? Would this same explanation hold true for flood events?

Reviewer #2: All comments are captured above. The authors miss a huge opportunity to communicate their findings and the implications of their study results. Unfortunately, this study report is overly technical and relies on jargon and would benefit from sharing the discussion text with colleagues who do not perform these software models routinely. I strongly urge the authors to revisit each figure and legend, the results section and the first paragraph of the discussion section. Consider whether the text could be clearer for the readership of this journal.

Reviewer #3: Dear Editor,

The manuscript titled "Identification of Environmental Determinants Involved in the Distribution of Burkholderia pseudomallei in Southeast Asia using MaxEnt software" investigates the environmental variables predicting the suitable ecological niche for Burkholderia pseudomallei (Bp), a bacterium causing melioidosis, in Southeast Asia. Using MaxEnt modeling combined with systematic reviews and meta-analysis, the study identifies bioclimatic and edaphic factors as key determinants of Bp's distribution. The most influential variables include the mean temperature of the wettest quarter (≥26°C), annual precipitation (<2300 mm), and Acrisol soil type. The study produces a distribution map highlighting areas with high suitability for Bp presence, significantly contributing to public health planning and disease burden assessment in the region under current climatic conditions. 

It is a well-written manuscript. While the manuscript seems thorough and scientifically robust, there are a few limitations that could be considered:

1. Geographic Scope: The study focuses specifically on Southeast Asia. While this is a strength in terms of depth, it may limit the applicability of the findings to other regions where Burkholderia pseudomallei is found, such as Northern Australia and other tropical areas. Extending the study to include these regions could provide a more comprehensive understanding of the environmental determinants.

Model Dependence: The study heavily relies on MaxEnt software for ecological niche modeling. While MaxEnt is a widely used and robust tool, the results are inherently dependent on the assumptions and limitations of the model, such as the reliance on presence-only data and the potential for overfitting, especially with a large number of predictors.

2. Data Availability: The manuscript mentions that all research data can be accessed upon request, which could limit immediate verification and replication of the study results by other researchers. Making data openly available in a recognized repository could enhance transparency and reproducibility.

3. Environmental Variables: The selection of environmental variables, while extensive, might still omit other potentially influential factors that could affect the distribution of Burkholderia pseudomallei. For instance, human activity or land-use changes, which are significant in tropical regions, could also impact the bacterium’s distribution but are not included in the model.

4. Static Time Frame: The environmental data used for the model are presumably static or represent average conditions. This does not account for year-to-year climatic variability or long-term climate change, which could significantly affect the distribution patterns of the pathogen.

5. Confirmation of Model Predictions: While the model predicts potential distributions based on environmental suitability, actual occurrences of Burkholderia pseudomallei would need to be confirmed through field sampling and laboratory confirmation, which could provide a robust test of the model's accuracy.

Addressing these limitations in future research could provide a more nuanced understanding of the factors influencing the distribution of Burkholderia pseudomallei and potentially improve the predictive capabilities of the models used.

PLOS authors have the option to publish the peer review history of their article (what does this mean?). If published, this will include your full peer review and any attached files.

Reviewer #1: No

Reviewer #2: No

Reviewer #3: No
---

## [Decision Letter · Decision Letter 1]

18 Sep 2024

Dear Dr Villarante Abrantes,

Thank you very much for submitting your manuscript "Identification of Environmental Determinants Involved in the Distribution of Burkholderia pseudomallei in Southeast Asia using MaxEnt software" for consideration at PLOS Neglected Tropical Diseases. As with all papers reviewed by the journal, your manuscript was reviewed by members of the editorial board and by several independent reviewers. The reviewers appreciated the attention to an important topic. Based on the reviews, we are likely to accept this manuscript for publication, providing that you modify the manuscript according to the review recommendations. 

Sincerely,

Dawit Gebeyehu Getachew, MPH

Guest Editor

Stuart Blacksell

Section Editor

Reviewer's Responses to Questions

**Key Review Criteria Required for Acceptance?**

**Methods**

-Are the objectives of the study clearly articulated with a clear testable hypothesis stated?

-Is the study design appropriate to address the stated objectives?

-Is the population clearly described and appropriate for the hypothesis being tested?

-Is the sample size sufficient to ensure adequate power to address the hypothesis being tested?

-Were correct statistical analysis used to support conclusions?

-Are there concerns about ethical or regulatory requirements being met?

Reviewer #1: (No Response)

Reviewer #3: Yes, to all. the revised manuscript has addressed most of the deficiencies.

**Results**

-Does the analysis presented match the analysis plan?

-Are the results clearly and completely presented?

-Are the figures (Tables, Images) of sufficient quality for clarity?

Reviewer #1: (No Response)

Reviewer #3: Yes.

**Conclusions**

-Are the conclusions supported by the data presented?

-Are the limitations of analysis clearly described?

-Do the authors discuss how these data can be helpful to advance our understanding of the topic under study?

-Is public health relevance addressed?

Reviewer #1: (No Response)

Reviewer #3: Yes

**Editorial and Data Presentation Modifications?**

Reviewer #1: (No Response)

Reviewer #3: (No Response)

**Summary and General Comments**

Reviewer #1: I believe the authors have responded appropriately to the majority of the reviewer comments.

I have included a few minor comments to help improve the readability of the manuscript.

Abstract:

The use of (Bio8) and (Bio12) in the abstract is somewhat unhelpful without prior explanation.

Line 260-262: it would be easier for the reader to interpret these sentences if the “Bio” included the associated factor just in this section when they are 1st mentioned in the main text e.g. Bio8 (mean temperature…) ; Biocat1 (soil type).

Line 276: cloglog, technically an abbreviation and should be spelled out on first use.

Line 340: Melioidosis doesn’t need to be capitalised

Line 386: check grammar/syntax

Line 438: remove “even”

Reviewer #3: Dear Editor,

The revised manuscript has been significantly improved and addressed most of the reviewers' comments and questions. However, I have a request that the authors should consider including their response to my concern #4 regarding the environmental variables below to the revised manuscript. This is a limitation of using the current methods and this limitation needs to be documented. 

"Environmental Variables: The selection of environmental variables, while extensive, 

might still omit other potentially influential factors that could affect the distribution of 

Burkholderia pseudomallei. For instance, human activity or land-use changes, which 

are significant in tropical regions, could also impact the bacterium’s distribution but are 

not included in the model.

Authors’ Response: For this study, we have only focused on the correlative 

nature of the MaxEnt modeling. Hence, we only utilized factors that we have 

easily accessed with such as the bioclimatic factors and topographic/edaphic 

factors. For a more mechanistic approach, another phase of the modeling study 

the authors intend to do. We have plans of exploring the potentialities of 

incorporating such anthropogenic data if available"

PLOS authors have the option to publish the peer review history of their article (what does this mean?). If published, this will include your full peer review and any attached files.

Reviewer #1: No

Reviewer #3: Yes: Apichai Tuanyok

Figure Files:

Data Requirements:

Reproducibility:

References

---

## [Editor Report · Decision Letter 2]

20 Oct 2024

Dear Dr. Jose Francis,

Thank you very much for submitting your manuscript "Identification of Environmental Determinants Involved in the Distribution of Burkholderia pseudomallei in Southeast Asia using MaxEnt software" for consideration at PLOS Neglected Tropical Diseases. As with all papers reviewed by the journal, your manuscript was reviewed by members of the editorial board and by several independent reviewers. The reviewers appreciated the attention to an important topic. Based on the reviews, we are likely to accept this manuscript for publication, providing that you modify the manuscript according to the review recommendations.

Sincerely,

Dawit Gebeyehu Getachew, MPH

Guest Editor

Stuart Blacksell

Section Editor

Figure Files:

Data Requirements:

Reproducibility:

References

---

## [Editor Report · Decision Letter 3]

7 Nov 2024

Dear Mr Abrantes,

We are pleased to inform you that your manuscript 'Identification of Environmental Determinants Involved in the Distribution of Burkholderia pseudomallei in Southeast Asia using MaxEnt software' has been provisionally accepted for publication in PLOS Neglected Tropical Diseases.

Best regards,

Stuart D. Blacksell

Section Editor

Shaden Kamhawi

co-Editor-in-Chief

Paul Brindley

co-Editor-in-Chief

---

## [Editor Report · Acceptance letter]

17 Dec 2024

Dear Mr Abrantes,

We are delighted to inform you that your manuscript, "Identification of Environmental Determinants Involved in the Distribution of Burkholderia pseudomallei in Southeast Asia using MaxEnt software," has been formally accepted for publication in PLOS Neglected Tropical Diseases.

Best regards,

Shaden Kamhawi

co-Editor-in-Chief

Paul Brindley

co-Editor-in-Chief
